# Relationships between the ground reaction force during initial sprint acceleration and the vertical force–velocity profile

**Motoki Katsuge**[1], **Hikaru Kurosaki**[1,2], **Hiromu Watanabe**[3], **Sohma Kambayashi**[1], **Kosuke Hirata**[4], **Kuniaki Hirayama**[2*]

**1** Graduate School of Sport Sciences, Waseda University, Tokorozawa, Japan, **2** Faculty of Sport Sciences, Waseda University, Tokorozawa, Japan, **3** School of Sport Sciences, Waseda University, Tokorozawa, Japan, **4** Institute of Health and Sport Sciences, University of Tsukuba, Tsukuba, Japan

* k.hirayama@waseda.jp

## Abstract

### Aim

This study examined the relationships between the ground reaction force (GRF) during sprint acceleration and lower-limb mechanical capabilities derived from the vertical force–velocity (F-V) profile.

### Materials and methods

Thirty-one male collegiate baseball players performed 15-m sprint accelerations. The mean horizontal and resultant GRFs and leg extension velocities in the propulsive phase were calculated for the first, fifth, and ninth steps during sprint acceleration. From the F-V profile estimated by squat jumps under 5–6 loading conditions (0–100 kg), the theoretical maximum force ($F_0$), velocity ($V_0$), power ($P_{max}$), and dynamic lower-limb strength corresponding to the leg extension velocities at each step during sprint acceleration ($F_{1st}$, $F_{5th}$, and $F_{9th}$) were obtained. Correlations between GRFs during sprint acceleration and F-V profile-derived variables were examined.

### Results

$F_0$ moderately to largely correlated with the horizontal GRFs for all steps ($r = .359$ to .543; $P = .002$ to .047). $P_{max}$ moderately correlated with the horizontal GRFs for the fifth and ninth steps ($r = .357$ and .448; $P = .049$ and .011, respectively) and resultant GRF for the ninth step ($r = .380$; $P = .035$). No significant correlations existed between dynamic lower-limb strengths and GRFs, except for $F_{1st}$ and resultant GRF for the first step ($r = .364$; $P = .045$).

**Data availability statement:** All relevant data are within the paper and its Supporting Information files.

**Funding:** This work was supported by JSPS KAKENHI Grant Number JP21K11427 to KH and a Waseda University Grant for Special Research Projects (project number: 2023C-539) to KH. The funders had no role in study design, data collection and analysis, decision to publish, or preparation of the manuscript.

**Competing interests:** The authors have declared that no competing interests exist.

## Conclusions

Greater lower-limb maximal strength and power contribute to a greater horizontal GRF generation in the entire and latter early phases of sprint acceleration, respectively. Thus, strength training tailored to neuromuscular demands for each step may be effective for enhancing sprint acceleration performance.

## Introduction

Sprint acceleration performance is essential for field sports athletes. Although the horizontal velocity of the athlete's center of mass continues to increase from the start of sprint acceleration until approximately 30 m [1], the mean distance of sprint activities is between 10 and 20 m during competitions in many field sports [2]. Hence, to enhance the sprint acceleration performance in field sports, the sprint acceleration in a short distance (i.e., initial sprint acceleration) must be improved.

Previous studies have reported that a faster 10-m sprint time correlated with the superior mechanical capabilities of lower-limb muscles such as higher one repetition maximum of back squat [3] and higher power outputs in vertical jumps [4]. To achieve higher velocity during sprint acceleration, greater horizontal ground reaction force (GRF) is required during the contact phase [5]. Thus, the superior mechanical capabilities of lower-limb muscles were assumed to lead to larger horizontal GRFs during sprint acceleration. Conversely, the force exerted on the ground is the resultant GRF, and the horizontal GRF is only one component of the resultant GRF [6]. That is, the magnitude of the horizontal GRF is not completely dependent on the magnitude of the resultant GRF because the ability to horizontally apply the GRF (i.e., the orientation of the resultant GRF vector) is considered one of the technical skills [7] that vary among individuals [8]. Therefore, the mechanical capabilities of lower-limb muscles may contribute more to the resultant GRF than to the horizontal GRF.

Capabilities to produce force and power during lower-limb extension depend on movement velocity and are well explained by the linear force–velocity (F-V) relationship [9]. The F-V relationship in the lower limbs is frequently assessed through vertical jumps under varied loads and summarized through three parameters: theoretical maximum force ($F_0$), velocity ($V_0$), and power ($P_{max}$) [10]. Previous studies [11–14] have investigated the associations with the $F_0$ of the squat jump (SJ) and the theoretical maximum horizontal GRF during sprint acceleration (i.e., the theoretical value of the horizontal GRF at zero velocity as extrapolated from the linear sprint F-V relationship). However, to the best of our knowledge, no studies have clarified the relationships between F-V parameters and the actual horizontal and resultant GRFs during sprint acceleration comprehensively. The force production is velocity-dependent, and greater force is generated at lower velocities, and vice versa [9]. The F-V characteristics vary individually ($F_0$, $V_0$, and slope of the F-V profile) [15,16]. For instance, individuals with identical $F_0$ but different slopes do not have the same force generation capacity at a given velocity. Thus, a deeper understanding of the contributions of the maximal mechanical capabilities of lower-limb muscles to the GRF during sprint

acceleration can be obtained by using individual F-V profiles, which allows for estimating of potential lower-limb strength at the leg extension velocity observed during sprint acceleration. As the GRF is the only source to horizontally accelerate the human body during sprint (when ignoring air resistance), an understanding of the physical attributes that determine the GRF during sprint acceleration is important to design training interventions aimed at improving sprint acceleration performance.

This study aimed to determine the relationships between the F-V profile obtained by SJs and the GRF during initial sprint acceleration. We hypothesized that (1) the $F_0$ would be positively correlated with horizontal and resultant GRFs during initial sprint acceleration and be more strongly associated with the resultant GRF than the horizontal GRF, and (2) the GRF during initial sprint acceleration would be positively correlated with the lower-limb strength in the leg extension velocity of the initial sprint acceleration calculated from the F-V profile, and the magnitude of the correlation would be greater than the relationships between the GRF and $F_0$.

## Materials and methods

### Participants

Through a priori power analysis, the sample size for the correlation analysis was computed using G*Power statistical power analysis software (G*Power 3.1.9.7; Kiel University, Germany). Based on a previous study [11], an effect size of 0.50 was assumed. The type 1 error and statistical power were set at 0.05 and 0.80, respectively. The critical sample size was computed to be 26, and 31 male collegiate baseball players were recruited from the same team competing at the top level of the Japanese college league (mean ± SD; age, 20 ± 1 years; height, 1.74 ± 0.06 m; body mass, 75.5 ± 12.0 kg), ensuring an adequate sample size. According to the classification established by McKay et al. [17], participants were classified as highly trained (Tier 3). No participants reported having neuromuscular diseases or musculoskeletal injuries specific to the lower limbs. Recruitment was started on 1 April 2022 and completed 30 April 2022.

Before participation, they were informed about the experimental procedure, potential risks, and the purpose of this study. All participants provided written informed consent. This study was approved by the Ethics Review Committee on Human Research of Waseda University (2021–345) and performed in accordance with the Declaration of Helsinki.

### Experimental design

In this cross-sectional study, participants performed a 10-min standardized warm-up consisting of jogging and lower-limb dynamic stretching, followed by SJs and sprint accelerations on the same day randomly. Participants avoided strenuous physical activity the day before the experiment. No familiarization session was provided because all participants were well-familiarized with jumping and sprinting based on the daily training and routine testing.

### Methodology

#### 15-m sprint acceleration

To measure the GRF during sprint acceleration, participants performed a total of six 15-m sprint accelerations from a standing start (Fig 1). Before the test, participants performed a specific warm-up consisting of three submaximal sprint accelerations over 15 m (one repetition each at 70%, 80%, and 90% of perceived maximal velocity). Participants started all tests from a two-point crouching position (staggered stance) with the right foot 0.5 m behind the starting line. They started sprinting at their command and ran as fast as possible. The GRFs for the first, fifth, and ninth steps were collected using a force plate (6012–15, Bertec Corporation, Columbus, USA) and in separate trials (two trials for each) by changing the starting line: the first two sprints were used to measure the first step, the second two sprints were collected for the fifth step, and the last two sprints for the ninth step. For each step measurement, the starting line was adjusted individually for each participant to ensure that they could naturally contact the force plate without altering their sprint motion. If the

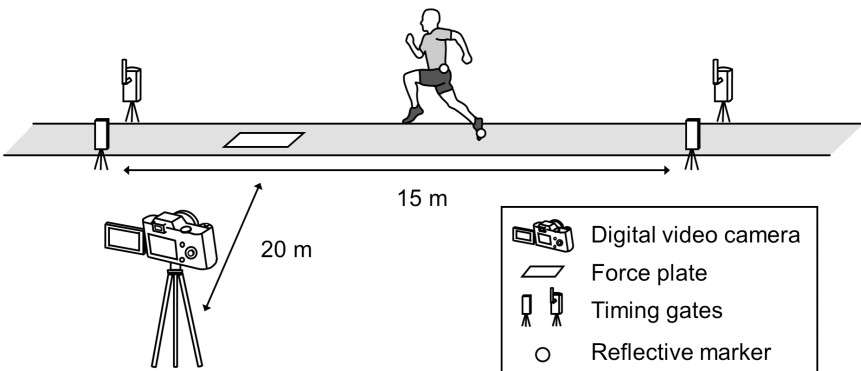

**Fig 1. Experimental setup of the 15-m sprint acceleration.**

participant did not make full foot contact on the force plate, adjustments were made to the starting line to ensure contact in subsequent trials. Two valid trials were performed for each step, with more than 2 min of recovery between trials.

Analog data obtained from the force plate were digitally converted using an analog-to-digital converter (Power-Lab/16SP, ADInstruments, Sydney, Australia) and collected using recording software (LabChart version 8, ADInstruments) at a sampling rate of 1,000 Hz. Reflective markers were placed on the left ilium (midpoint of the line segment connecting the left anterior superior iliac spine and left posterior superior iliac spine) and the left distal fifth metatarsal. Sprint acceleration motion was recorded from a sagittal plane at 240 Hz using a digital video camera (DMC-FZ300, Panasonic, Osaka, Japan). The camera was positioned 20 m away from the force plate and perpendicular to the sagittal plane. Force plate data and videos were synchronized using a synchronization device (FPF-AD02, 4Asist, Tokyo, Japan). The sprint time was measured using dual-beam electronic timing gates (Speedlight Timing System, Swift Performance, Queensland, Australia).

Touchdown and takeoff from the force plate were defined as when the vertical GRF first rose above 20 N and declined below 20 N, respectively [18]. The phase with a positive horizontal GRF was defined as the propulsive phase. The mean horizontal and resultant GRFs during the propulsive phase were calculated. All mean GRFs were normalized to the body mass. The mean angle of the GRF vector during the propulsive phase was calculated as follows:

$$\theta_{GRF} = \tan^{-1}\left(\frac{GRF_v}{GRF_H}\right) \cdot \frac{180}{\pi}$$

(1)

where $\theta_{GRF}$ is the mean angle of the GRF vector and $GRF_V$ and $GRF_H$ are the mean vertical and horizontal GRFs during the propulsive phase, respectively. Reflective markers were digitized using video analysis software (Frame-DIAS 6, Q'sfix, Tokyo, Japan) to determine the leg extension velocity during sprint acceleration. The line segment connecting the two markers was defined as the leg length, and the mean leg extension velocities for the first, fifth, and ninth steps during sprint acceleration were calculated by dividing the difference in the leg length at the start of the propulsive phase and the time of the takeoff by the time of the propulsive phase. The fastest trial was analyzed for each step.

## Squat jumps (SJs)

To determine the vertical F-V profile, participants performed SJs without external loads and against four external loads (20, 40, 60, and 80 kg), starting with the lightest load (Fig 2). Participants who could jump 80 kg using the proper technique performed additional 100-kg jumps after completing the 80-kg condition. Before the test, participants performed a back squat protocol consisting of three repetitions at 20 kg, three repetitions at the weight of half the body mass, three

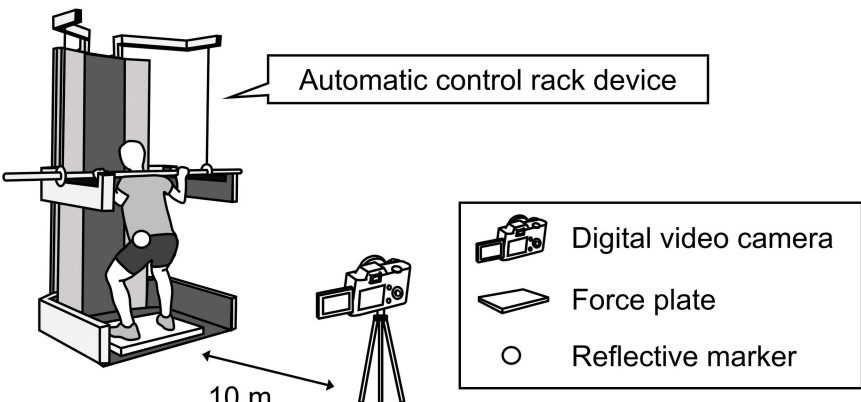

**Fig 2. Experimental setup of the squat jump.**

repetitions at the weight of the body mass, and an SJ protocol consisting of three unloaded SJs and three SJs loaded with 20 kg. An automatic control rack device (LIFTER, Intelligent Motion gmbh, Wartberg an der Krems, Austria) was used to reduce the landing load and anxiety, which possibly prevented performing a maximal effort jump. Before the measurement, the height from the barbell placed on the participant's shoulders to the ground, with knee and hip joints at 90°, was measured. Based on this height, the starting height of the rack-device safety bar was determined for each participant. The participants were instructed to hold the wooden rod (unloaded condition) or the free-weight barbell (loaded condition) on their shoulders, maintain the starting position with the knee and hip joints at 90°, and then jump as explosive and high as possible. Participants were also asked not to make countermovements before the start of the jump and not to take the bar off their shoulders until they reached the highest point of the jump. If these requirements were not met, the trial was repeated. Two valid trials were performed for each load, with 1 min recovery between trials and 3 min recovery between different loads.

SJs were performed with the participants standing on a force plate (Type 9281E, Kistler, Winterthur, Switzerland). SJ motion was recorded from 10 m behind the participant using a digital video camera (as mentioned above). The conversion and collection of analog data collected from the force plate and reflective marker conditions were identical to those for the 15-m sprint acceleration. Force plate data and videos were synchronized using a synchronization device (PH-145, Q'sfix).

Jump initiation was defined as the point at which the vertical GRF first rose >20 N of the system weight (body weight + weight of the external load), and takeoff was identified as that which the vertical GRF fell <20 N. The net vertical GRF was calculated as the amount of force exceeding the system weight and then divided by the system mass to determine the vertical acceleration of the system center of mass. The vertical velocity of the system center of mass was determined by integrating acceleration with respect to time using the trapezoidal role, and jump height was obtained from the vertical velocity at takeoff [19]. The push-off phase referred to the period from jump initiation to takeoff. The mean vertical GRF of the highest jump under each loading condition was obtained from the instantaneous values recorded over the entire push-off phase. All mean GRFs were normalized to the body mass. To determine the leg extension velocity during an SJ, reflective markers of the highest jump in each loading condition were digitized using video analysis software (as mentioned above). The displacement of the marker from jump initiation to takeoff was obtained from the coordinate values obtained by digitizing the reflective marker attached to the left ilium. The mean vertical leg extension velocity during an SJ was calculated by dividing the displacement of the marker by the time of the push-off phase.

The force and velocity for each loading condition were modeled by least-squares linear regressions to determine the individual F-V profile (Fig 3A):

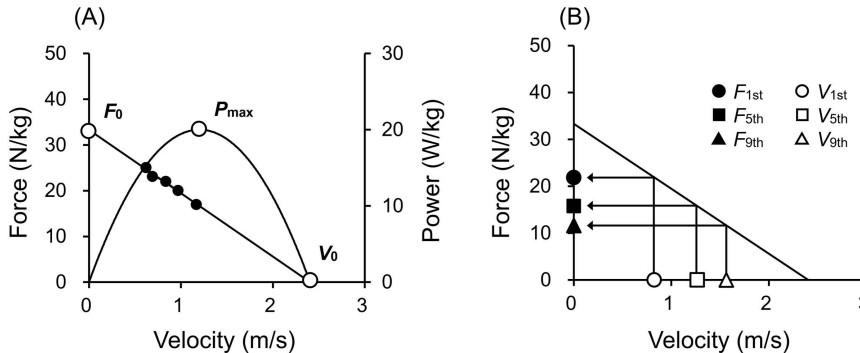

**Fig 3. (A) Typical example of a force–velocity (F-V) profile obtained from squat jumps. Each closed circle plot represents raw mean force and velocity during a squat jump at a given load. The F-V relationship for this participant was $F = F_0 - \alpha V$, where $F_0$ (theoretical maximum force) was 38.0 N/kg and $\alpha$ was 14.6 Ns/m. $P_{max}$ (24.8 W/kg), theoretical maximum power; $V_0$ (2.60 m/s), theoretical maximum velocity. (B) Typical example of calculation of dynamic lower-limb strength ($F_{1st}$, $F_{5th}$, and $F_{9th}$) corresponding to the mean leg extension velocities for the first, fifth, and ninth steps ($V_{1st}$, $V_{5th}$, and $V_{9th}$) during sprint acceleration. The $F_{1st}$ (24.4 N/kg) for this participant was calculated by substituting the $V_{1st}$ (0.93 m/s) for the F-V relationship. Similarly, the $F_{5th}$ (19.7 N/kg) and $F_{9th}$ (18.8 N/kg) were obtained using the values of $V_{5th}$ (1.26 m/s) and $V_{9th}$ (1.31 m/s).**

$$F = F_0 - \alpha V \tag{2}$$

where $F$ and $V$ represents the mean vertical GRF and leg extension velocity during a SJ at a given load, $F_0$ represents the theoretical maximum force (force–axis intercept), and $\alpha$ is the slope of the linear F-V relationship. Because the F-V relationship was highly linear, the theoretical maximum power ($P_{max}$) was determined as:

$$P_{max} = F_0 \times V_0/4 \tag{3}$$

where $V_0$ represents the theoretical maximum velocity (velocity–axis intercept) [9].

### Dynamic lower-limb strength

As a dynamic lower-limb strength, the mean GRF in the SJ corresponding to the mean leg extension velocity for the first step during sprint acceleration ($F_{1st}$) was calculated for each participant using the linear regression equation of the F-V profile (equation 2):

$$F_{1st} = \alpha V_{1st} + F_0 \tag{4}$$

where $\alpha$ is the regression coefficient; $V_{1st}$, the mean leg extension velocity for the first step during sprint acceleration; and $F_0$, the theoretical maximum force. The mean GRFs in the SJ corresponding to the mean leg extension velocities for the fifth and ninth steps during sprint acceleration ($F_{5th}$ and $F_{9th}$) were also calculated (Fig 3B).

### Statistical analyses

All results are presented as means ± SDs (Table 1). Data normality was analyzed using the Shapiro–Wilk test. Pearson's product–moment correlation coefficients with 95% confidence intervals were calculated to examine the relationships between GRFs during 15-m sprint acceleration and $F_0$, $V_0$, $P_{max}$, $F_{1st}$, $F_{5th}$, $F_{9th}$, and angles of the GRF vector. The threshold values for the interpretation of the correlation coefficient as an effect were as follows: < 0.1 (negligible), 0.1–0.3 (small),

**Table 1. Descriptive data of sprint-related variables and parameters derived from the force–velocity profile.**

| | | Mean ± SD |
|---|---|---|
| **Sprint-related variables** | | |
| Mean horizontal GRF | First step, N/kg | 5.38 ± 0.58 |
| | Fifth step, N/kg | 4.20 ± 0.46 |
| | Ninth step, N/kg | 3.92 ± 0.41 |
| Mean resultant GRF | First step, N/kg | 13.24 ± 0.98 |
| | Fifth step, N/kg | 16.19 ± 1.09 |
| | Ninth step, N/kg | 15.71 ± 1.62 |
| Mean angle of the resultant GRF vector | First step, degree | 65.99 ± 2.00 |
| | Fifth step, degree | 74.73 ± 1.38 |
| | Ninth step, degree | 75.23 ± 1.41 |
| Mean leg extension velocity | First step, m/s | 0.89 ± 0.16 |
| | Fifth step, m/s | 1.02 ± 0.30 |
| | Ninth step, m/s | 1.45 ± 0.24 |
| **Parameters derived from the F-V profile** | | |
| F-V variable | $F_0$, N/kg | 35.83 ± 3.88 |
| | $V_0$, m/s | 2.37 ± 0.33 |
| | $P_{max}$, W/kg | 21.04 ± 2.75 |
| Dynamic lower-limb strength | $F_{1st}$, N/kg | 21.90 ± 2.45 |
| | $F_{5th}$, N/kg | 19.87 ± 3.94 |
| | $F_{9th}$, N/kg | 13.13 ± 4.55 |

Abbreviations: F-V, force-velocity; $F_0$, theoretical maximum force; $F_{1st}$, $F_{5th}$, and $F_{9th}$, mean ground reaction forces in squat jump corresponding to the mean leg extension velocity for the first, fifth, and ninth steps during sprint acceleration, respectively; GRF, ground reaction force; $P_{max}$, theoretical maximum power; SD, standard deviation; $V_0$, theoretical maximum velocity.

0.3–0.5 (moderate), 0.5–0.7 (large), 0.7–0.9 (very large), and >0.9 (extremely large) [20]. The significance level was set at 0.05. All statistical analyses were performed using JASP version 0.16.2 (University of Amsterdam, The Netherlands).

## Results

Table 1 depicts the descriptive data for variables related to sprint acceleration, F-V profile parameters, and dynamic lower-limb strength. The correlation coefficients of the F-V profile parameters and dynamic lower-limb strength with the GRF during sprint acceleration are presented in Figs 4 and 5.

$F_0$ presented moderate to large significant correlations with the mean horizontal GRFs for all steps ($r$=.359 to .543; $P$=.002 to .047). $P_{max}$ presented moderate significant correlations with the mean horizontal GRFs for the fifth and ninth steps ($r$=.357 and .448; $P$=.049 and .011, respectively) and the mean resultant GRF for the ninth step ($r$=.380; $P$=.035). No significant correlations were found between $V_0$ and the horizontal/resultant GRFs for all steps ($r$=−.299 to .149; $P$=.215 to .852). $F_0$ moderately and significantly correlated with mean angles of the GRF vector for the first ($r$=−.477; $P$=.007) and fifth ($r$=−.402; $P$=.025) steps but not with the ninth step ($r$=−.059; $P$=.753).

$F_{1st}$ moderately correlated with the mean resultant GRF for the first step ($r$=.364; $P$=.045). No other significant correlations were observed between dynamic lower-limb strengths and GRFs ($r$=−.291 to .306; $P$=.094 to .761).

## Discussion

To the best of our knowledge, this study is the first to explore the underlying lower-limb mechanical capabilities of the GRF for several steps during initial sprint acceleration. A higher $F_0$ was found to be associated with a larger horizontal GRF for

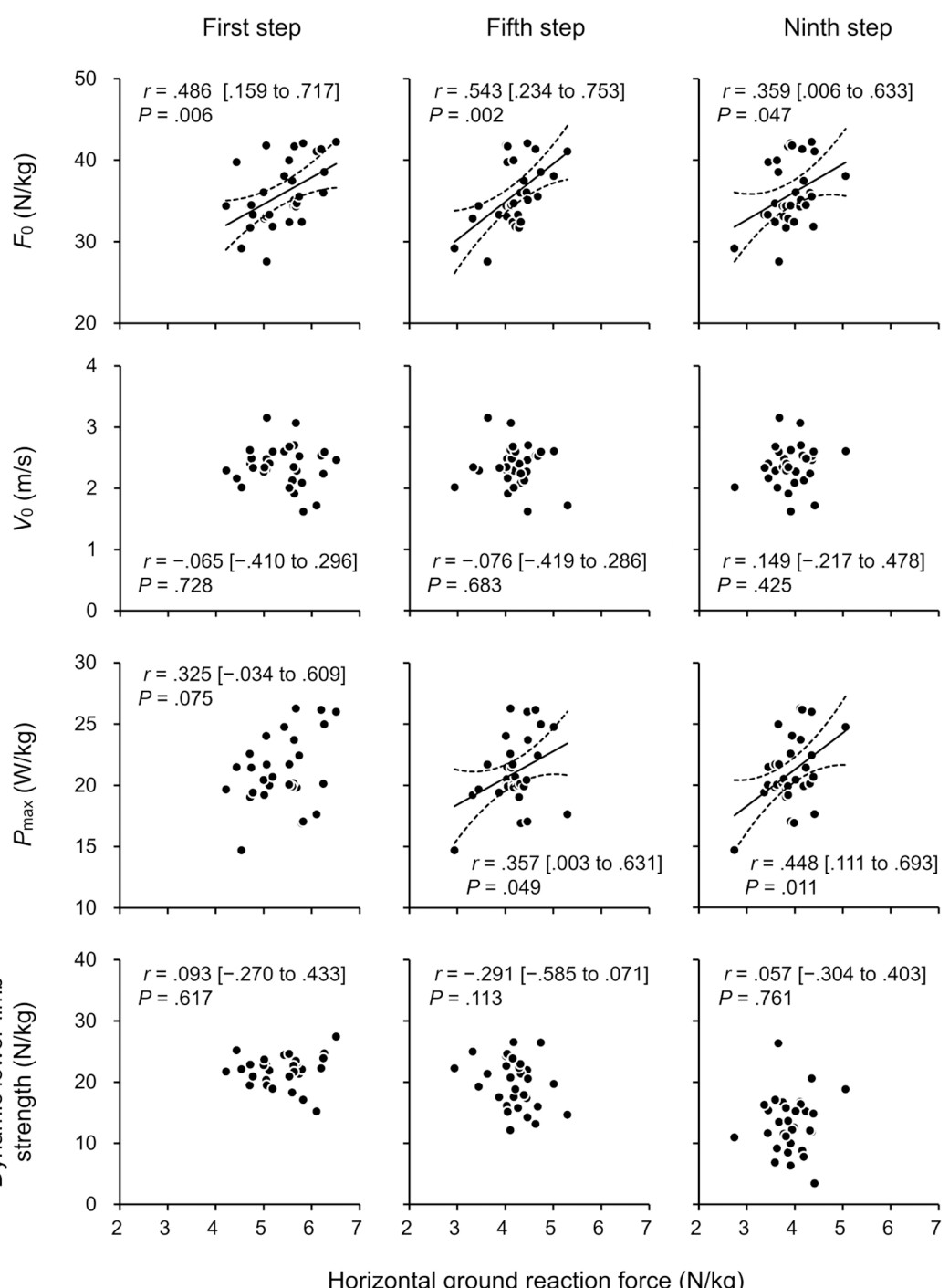

**Fig 4. Correlations (95% confidence intervals) of the horizontal ground reaction force for the first, fifth, and ninth steps (left, center, and right columns, respectively) during 15-m sprint acceleration with parameters of the force–velocity profile and dynamic lower-limb strength.** $F_0$, theoretical maximum force; $P_{max}$, theoretical maximum power; $V_0$, theoretical maximum velocity.

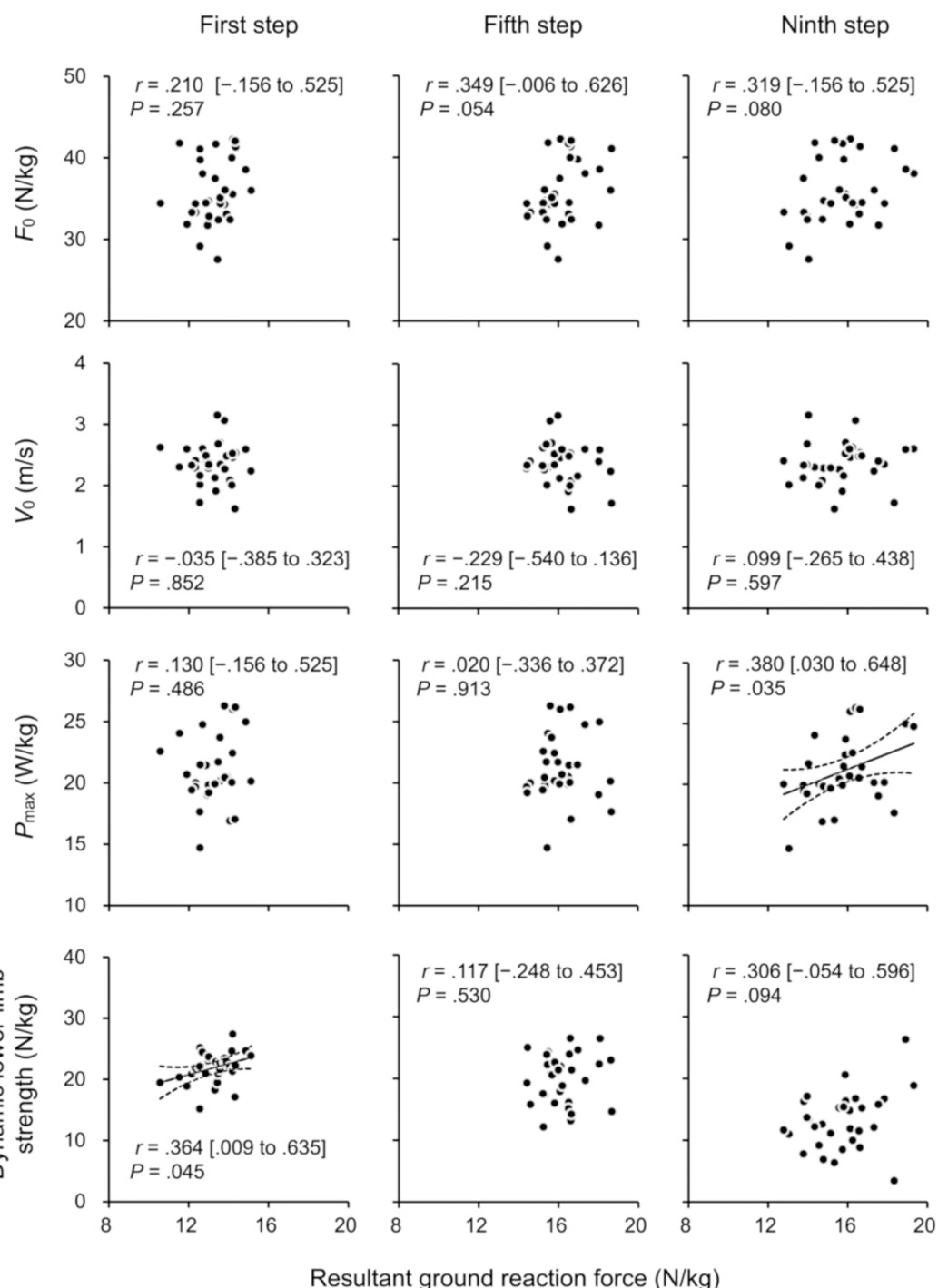

**Fig 5. Correlations (95% confidence intervals) of the resultant ground reaction force for the first, fifth, and ninth steps (left, center, and right columns, respectively) during 15-m sprint acceleration with parameters of the force–velocity profile and dynamic lower-limb strength.** $F_0$, theoretical maximum force; $P_{max}$, theoretical maximum power; $V_0$, theoretical maximum velocity.

the first, fifth, and ninth steps during sprint acceleration but not with the resultant GRF. This finding did not fully support the first hypothesis that the horizontal and resultant GRFs would correlate with $F_0$ and the magnitude of the correlation would be greater for the resultant GRF. Moreover, no significant correlations were found between the GRF and dynamic lower-limb strength except for the resultant GRF for the first step, so the second hypothesis was rejected, i.e., the GRF during initial sprint acceleration would positively correlate with the lower-limb strength in the leg extension velocity of the initial sprint acceleration calculated from the F-V profile.

A higher $F_0$ significantly correlated with larger horizontal GRFs for the first, fifth, and ninth steps during sprint acceleration. Theoretically, the amount of the horizontal GRF applied onto the ground is determined by both the magnitude and vector orientation of the resultant GRF. In this study, greater $F_0$ was also associated with a more forward inclination of the resultant GRF vectors for the first and fifth steps, but did not significantly correlate with the magnitude of the resultant GRF. These results imply that the lower-limb maximal strength may influence the direction of the resultant GRF vector rather than its magnitude. During sprint acceleration, a lower resultant GRF might be beneficial for the effective generation of a horizontal GRF, thereby resulting in better sprint acceleration performance [21]. Kugler and Janshen [21] observed that individuals with superior acceleration performance trended toward lower resultant GRF and significantly longer ground contact times, leading to a greater forward lean of the body and therefore greater horizontal force application, particularly during the latter contact phase. In addition, a previous study [22] indicated that during sprint acceleration, a greater increase in sprint velocity was achieved with a small vertical GRF, which is a primary determinant of the magnitude of the resultant GRF, since the magnitude of the vertical GRF required during sprint acceleration might be the one that creates a flight time only long enough to reposition the lower limbs [23]. Therefore, individuals with greater lower-limb strength are likely able to tilt the GRF vector forward while maintaining a certain amount of vertical GRF, and greater lower-limb strength does not necessarily contribute to an increased magnitude of the resultant GRF.

Since producing a large force in a fast movement is required to increase sprint velocity, the ability to generate high power and velocity are considered important physical parameters [24]. This study showed that $P_{max}$ correlated with the horizontal GRF for the fifth and ninth steps. Previous studies [25,26] have indicated that lower-limb power was associated with the latter phase of the initial sprint acceleration but not with the earlier phase. For example, Chelly et al. [25] demonstrated that the average power per body mass during SJ without external load was significantly correlated with the average sprint velocity over the first 5 m but not with that over the first step during 10-m sprint running. Furthermore, Nagahara et al. [26] reported that SJ jump height without external load was significantly correlated with acceleration from the sixth to the tenth steps, but not from the first to fifth steps during 60-m sprint running. Therefore, it can be speculated that the lower-limb capability to produce higher power contributes to producing a larger horizontal force during the latter steps, but not the earlier steps, of the initial sprint acceleration. In contrast to $P_{max}$, no correlation was observed between $V_0$ and the horizontal and resultant GRFs for all steps. Morales-Artacho et al. [27] reported that isometric knee extensor torque and the rate of torque development were strongly related to $F_0$ and $P_{max}$ in the F-V profile obtained from countermovement jump, while there were no associations with $V_0$. Although the jump types used in Morales-Artacho et al. [27] and the current study differed, it is possible that $V_0$ in vertical jump may not adequately reflect intrinsic force-generating capabilities of the lower-limb, which may explain the lack of significant correlations with GRFs during sprint acceleration. Our results suggest that the ability to produce larger force and power is more important than the ability to produce higher velocity in the initial sprint acceleration.

The $P_{max}$ also exhibited a significant correlation with the mean resultant GRF for the ninth step. A previous study reported that during the 5–10 m interval of the 10-m sprint, the greater peak vertical GRF during the contact phase was correlated with a higher mean sprint velocity [28]. Considering that the power production capability of the lower-limb assessed by the SJ contributes to the greater sprint acceleration during the latter phase of the initial sprint acceleration [26] and that the magnitude of the resultant GRF is mainly determined by the vertical GRF, the maximal power of the lower-limb may also play an important role in producing a larger resultant GRF during the latter phase of the initial sprint acceleration.

Although $F_{1st}$ correlated with the resultant GRF for the first step, no other correlations between leg extension strength corresponding leg extension velocity during sprint acceleration and the GRF was found. This result implies that estimating the dynamic lower-limb strength for sprint acceleration from the F-V profile obtained from an SJ is not possible. In this study, the dynamic lower-limb strength was measured in the SJ, which does not include the stretch-shortening cycle (SSC) of the muscle-tendon units (MTUs). During sprint acceleration, the MTU's contraction pattern demonstrated an SSC, where MTUs initially elongated during the braking phase and then shortened in the subsequent propulsive phase [29]. In contrast, SJs involve mainly concentric contractions, representing force exertion without the SSC [30]. In the study, the resultant GRF for the ninth step was larger than the $F_{9th}$ for most participants, suggesting the effect of the SSC on increasing force production during sprint acceleration. In addition to the MTUs' contraction behavior, motor control strategies differ between the SJ and sprint acceleration. Both exercises demonstrate an identical proximal-to-distal sequence pattern of electromyographic activity in the lower-limb muscles, where the hip extensors are activated first, followed by the knee extensors, and finally the plantar flexors [31,32]. During sprint acceleration, however, the hamstrings and rectus femoris exhibit a more distinct reciprocal activation pattern than that observed during jumping [32]. These differences in MTUs' contraction behavior and motor control strategies between sprint acceleration and the SJ might explain the absence of a significant correlation.

Although it is known that phase-specific demands for kinetic characteristics and strength-power capabilities exist in sprint acceleration [22,26], the physical factors responsible for the sprint acceleration kinetics during specific steps are not clear. The present study revealed the phase-specific demands of physical factors for greater GRFs during sprint acceleration, implying that training interventions tailored to different steps are vital for effectively improving the initial sprint acceleration performance. The present findings imply that training interventions tailored to steps during sprint acceleration are important to effectively improve sprint acceleration performance. A study demonstrated that high-load strength training was effective in enhancing lower-limb maximal strength, and light-load explosive training increased maximal muscle power [33]. Thus, heavy-load strength training (e.g., back squats with higher load and lower repetitions) is possibly effective in increasing force production in the earlier phase of the initial sprint acceleration, and explosive strength training (e.g., jump squats with a lighter load) is effective in increasing force production in the later phase.

This study has some limitations. First, the study participants were only male baseball players. It has been suggested that several differences in sprint acceleration kinetics exist between different cohorts, such as those in sport [34], sex [35], and age [36]. Thus, the results might differ if athletes from other sports, female athletes, or youth and master athletes were employed, which warrants further investigation. Second, because this is a cross-sectional study, causal relationships remain unclear. Thus, further prospective research is needed to determine whether training-induced changes in the F-V profile can influence GRFs during initial sprint acceleration.

## Conclusions

This study revealed that greater maximal lower-limb muscle strength and power contribute to greater production of horizontal GRFs in the entire and later phases of initial sprint acceleration, respectively. These results may help coaches design strength and conditioning programs to improve specific mechanical capabilities that influence the force production in each step during initial sprint acceleration, thereby effectively enhancing sprint acceleration performance.

## Supporting information

**S1 Table. Original data.**
(XLSX)

## Acknowledgments

The authors would like to thank all the athletes who participated in this study.

## Author contributions

**Conceptualization:** Motoki Katsuge, Kuniaki Hirayama.

**Data curation:** Motoki Katsuge, Kuniaki Hirayama.

**Formal analysis:** Motoki Katsuge.

**Funding acquisition:** Kuniaki Hirayama.

**Investigation:** Motoki Katsuge, Hikaru Kurosaki, Hiromu Watanabe, Sohma Kambayashi, Kosuke Hirata.

**Methodology:** Motoki Katsuge, Kuniaki Hirayama.

**Project administration:** Kuniaki Hirayama.

**Resources:** Kuniaki Hirayama.

**Software:** Motoki Katsuge.

**Supervision:** Kuniaki Hirayama.

**Validation:** Motoki Katsuge.

**Visualization:** Motoki Katsuge.

**Writing – original draft:** Motoki Katsuge.

**Writing – review & editing:** Motoki Katsuge, Kosuke Hirata, Kuniaki Hirayama.

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
