## [Decision Letter · Decision Letter 0]

Dear Dr. Hirayama,

Thank you for submitting your manuscript to PLOS ONE. After careful consideration, we feel that it has merit but does not fully meet PLOS ONE’s publication criteria as it currently stands. Therefore, we invite you to submit a revised version of the manuscript that addresses the points raised during the review process.

**ACADEMIC EDITOR: ** The authors need to address the major issues highlighted by both reviewers. The quality of the paper and the strength of the work need to be emphasized.

We look forward to receiving your revised manuscript.

Kind regards,

Andrea Tigrini, Ph.D.

Academic Editor

PLOS ONE

Journal Requirements:

1. When submitting your revision, we need you to address these additional requirements. Please ensure that your manuscript meets PLOS ONE's style requirements, including those for file naming. The PLOS ONE style templates can be found at https://journals.plos.org/plosone/s/file?id=wjVg/PLOSOne_formatting_sample_main_body.pdf and https://journals.plos.org/plosone/s/file?id=ba62/PLOSOne_formatting_sample_title_authors_affiliations.pdf

Additional Editor Comments:

The authors need to address the major issues highlighted by both reviewers. The quality of the paper and the strength of the work need to be emphasized.

Reviewers' comments:

Reviewer's Responses to Questions

**Comments to the Author**

1. Is the manuscript technically sound, and do the data support the conclusions?

Reviewer #1: Yes

Reviewer #2: Yes

2. Has the statistical analysis been performed appropriately and rigorously?

Reviewer #1: Yes

Reviewer #2: No

3. Have the authors made all data underlying the findings in their manuscript fully available?

Reviewer #1: Yes

Reviewer #2: Yes

4. Is the manuscript presented in an intelligible fashion and written in standard English?

Reviewer #1: Yes

Reviewer #2: Yes

Reviewer #1: Authors have done good work in understanding the relationships between the ground reaction force and lower limb mechanical capabilities (F-V profile parameters) for better designing of training interventions in order to protect athletes. However, some issues were observed while reading the manuscript as follow:

- In the methods section, it’s not clear if the reflective markers were captured using an ordinary digital camera? Was it equipped with infrared sensors?

- The final sampling rate after passing through the synchronizer is not declared. Also the use of hardware for resynchronization and not through processing is not justified.

- A figure for the experimental setup in page 9 might be useful for clarity of the protocol applied with the participants.

- Some info in lines 172-175 are redundant with previous pages.

- In the F(V) formula at page 10, F should actually a function of V which is not present in the terms, could authors please elaborate better in this equation of F-V relationship.

- Line 214 “ All data…..” must be all results.

- My principal concern relates to the novelty of the study and the clinical value that can be translated for real athletes. Although authors have declared that this is an important step in that direction, however the did not yet explain how can these findings specifically affect some sports individuals performance and training?

- Lines 286-294 : the interpretation of the lower limb strength is a bit confusing and need to be slightly modified for better clarity.

- The age of the cohort is very limited. I would like to understand the authors perspective about how these results can extend to any other athlete ages.

Reviewer #2: This paper examines the relationship between the force capacity of the lower limb, estimated by the F-V profile estimation method, and GRF during initial sprint acceleration. Although the paper is novel, there are some parts that seem insufficient in terms of clarity, and there are also some points that require compliance with a policy of self-plagiarism.

Comment as follows

- Perhaps, this paper is a series with the abstracts in following proceedings. Since the statistics in the abstract of this paper are almost the same as those listed in the proceedings below, we assume that the main issues are similar. Since I believe there is a discussion accompanying these statistical results in this paper, I assume there are differences from the following proceedings abstract. However, you should cite the following proceedings abstract in this paper and, in accordance with PLOS ONE policy, if necessary I think it is necessary to clarify the difference points.

- "Relationship between leg extension strength characteristics and ground reaction force during sprint acceleration," Motoki Katsuge, Kosuke Hirata, Hikaru Kurosaki, Hiromu Watanabe, Sohma Kambayashi, Kuniaki Hirayama, 2022 SESNZ Conference

- L152: You described “4–5 external loads (range, 20–100 kg).” The way this is written, this means that one load is 80 kg, and each of the four loads is 5 kg. I recommend that this be corrected to correctly convey the experimental conditions.

- L200: Fig 1 (A) clearly states that the figure is a typical example, is Fig 1 (B) also a typical example?

- L253: Have you mentioned the following results in your discussion? “the mean resultant GRF for the ninth step (r = .380; P = .035).”

- L298-299: “A plausible explanation for these results is the consistency of the leg extension speed during sprint acceleration and SJ. “ Perhaps it is due to my lack of understanding, but I could not understand your description. What is “consistency”?

- L299-304: Perhaps it is due to my lack of understanding, but I could not understand your logic. Why can you say that knee movements in a velocity band slower than the unloaded velocity (V0), the slowest of which is no longer related to GRF?

- L301-304: Are “similar” and “much smaller” statistically significant? I encourage you to discuss this statistically.

- L316-317: You have described the differences in the modes of contraction of the muscles. In addition to this, SJ and sprint have different motor control strategies, so the way the muscles are used should also be different. I think it would be better to mention this factor as well.

- In some cases, the same thing, such as speed or velocity, is expressed in different phrases. It is recommended to unify them.

**Do you want your identity to be public for this peer review?** For information about this choice, including consent withdrawal, please see our Privacy Policy

Reviewer #1: No

Reviewer #2: No

---

## [Author Response · Author response to Decision Letter 1]

4 Feb 2025

Dear Editor and Reviewers,

First of all, we would like to thank you for significantly extending the deadline for our response. Your generosity allowed us to enough time to consider the revisions. We also want to express our gratitude for the opportunity to improve our manuscript and for the reviewers’ thorough review and constructive suggestions. All comments have been carefully considered, and greatly contributed to enhancing the overall quality of our manuscript.

We have highlighted our responses to all points in blue. Additionally, the revisions to the manuscripts have been emphasized with bold text and underlines.

We hope our responses and revisions meet your expectations.

Yours faithfully,

Authors and co-authors

Additional Editor Comments

The authors need to address the major issues highlighted by both reviewers. The quality of the paper and the strength of the work need to be emphasized.

Response: We would like to thank the editor for the time and effort put into the reviewing process of our manuscript and for the opportunity to resubmit the revised version. We have addressed all points raised by the reviewers with great care.

Review Comments to the Author

Reviewer #1

Authors have done good work in understanding the relationships between the ground reaction force and lower limb mechanical capabilities (F-V profile parameters) for better designing of training interventions in order to protect athletes. However, some issues were observed while reading the manuscript as follow:

- In the methods section, it’s not clear if the reflective markers were captured using an ordinary digital camera? Was it equipped with infrared sensors?

Response: We used ordinary digital video camera, and it was not equipped with infrared sensors. We have now included this information in the method section (Track changes version: Lines 136 and 179; Original version: Lines 132 and 172)

- The final sampling rate after passing through the synchronizer is not declared. Also the use of hardware for resynchronization and not through processing is not justified.

Response: In this study, the final sampling rates for both the GRF and video data are the same as their respective original acquisition sampling rates (i.e., 1,000 Hz for the GRF data and 240 Hz for the video data). This approach was chosen because resampling the video data to match the GRF data is not feasible due to the fundamental differences in their original sampling rates. While we fully acknowledge the importance of aligning final sampling rates, we believe that this discrepancy does not compromise the validity of our findings. To explain the reason for this, we refer to the analysis of the first step in a 15-m sprint acceleration by participant 1 as an example (Fig below).

First, the GRF data and the video data were synchronized based on the activation point of the synchronizer. Then, the time intervals from this signal to the onset of the propulsive phase and to takeoff ((A) and (B), respectively) were calculated in the GRF data. Using these time points, we identified the corresponding time point in the video data. In cases where there was a mismatch between the GRF data and the video data, the video time point closest to the GRF-derived time points was selected.

Fig. A typical example of the detection of the onset of the propulsive phase and takeoff time for the first step during 15-m sprint acceleration.

(A) The time interval from this signal to the onset of the propulsive phase in the GRF data. (B) the time interval from this signal to the takeoff time. (C) The propulsive phase detected from the GRF data. The onset time of the propulsive phase and the takeoff time are 0.2540 s and 0.4620 s after the signal of the synchronizer. (D) The propulsive phase detected from the video data. The onset time of the propulsive phase and the takeoff time are 0.2541 s and 0.4583 s after the signal of the synchronizer.

To examine the potential impact of discrepancies in analysis intervals, we calculated the leg extension velocity for the first step during 15-m sprint acceleration under three conditions for 10 participants (Participants 1–10): (i) the interval identified as described above, (ii) the interval shifted one frame earlier than (i), and (iii) the interval shifted one frame later than (i).

The leg extension velocities for each condition were as follows: (i) 0.83 ± 0.18 m/s, (ii) 0.82 ± 0.16 m/s, and (iii) 0.83 ± 0.17 m/s. A one-way ANOVA was conducted to assess differences in leg extension velocity among the three conditions, and no significant differences were observed (p = 0.981). Furthermore, the magnitude of differences between conditions was examined using Cohen’s d, revealing negligible effect sizes (1 vs. 2: d = 0.08, 1 vs. 3: d = 0.01, 2 vs. 3: d = -0.07). These results indicate that the impact of frame misalignment on the calculation of leg extension velocity is negligible.

Therefore, we believe that analyzing the data at the original sampling rates does not lead to any considerable difference in the results. We hope these explanations adequately address your concerns and demonstrates the validity of our methodology.

- A figure for the experimental setup in page 9 might be useful for clarity of the protocol applied with the participants.

Response: We have added figures of the experimental setup for 15-m sprint acceleration and squat jump as Fig 1 and 2, respectively.

“Fig 1. Experimental setup of the 15-m sprint acceleration.” (Track changes version: Line 128)

“Fig 2. Experimental setup of squat jump.” (Track changes version: Line 175)

- Some info in lines 172-175 are redundant with previous pages.

Response: We have changed the part of the method section for SJ which was identical with the method of 15-m sprint acceleration (Track changes version: Lines 179-181; Original version: Lines 169-173):

“SJs were performed with the participants standing on a force plate (Type 9281E, Kistler, Winterthur, Switzerland). SJ motion was recorded from 10 m behind the participant using a digital video camera (as mentioned above). The conversion and collection of analog data collected from the force plate and reflective marker conditions were identical to those for the 15-m sprint acceleration. Force plate data and videos were synchronized using a synchronization device (PH-145, Q’sfix).”

- In the F(V) formula at page 10, F should actually a function of V which is not present in the terms, could authors please elaborate better in this equation of F-V relationship.

Response: As you pointed out, F (i.e., the mean vertical GRF during a SJ at a given load) is a function of V (i.e., the mean vertical leg extension velocity during a SJ at a given load). Therefore, “F(V) = F0 – αV0” in the original version is incorrect, and the correct formula is ”F = F0 – αV”. The manuscript has been corrected accordingly (Track changes version: Lines 199-204; Original version: Lines 191-195):

“The force and velocity for each loading condition were modeled by least-squares linear regressions to determine the individual F-V profile (Fig 3A): F = F0 − αV, where F and V represents the mean vertical GRF and leg extension velocity during a SJ at a given load, F0 represents the theoretical maximum force (force–axis intercept), and α is the slope of the linear F-V relationship. Because the F-V relationship was highly linear, the theoretical maximum power (Pmax) was determined as Pmax = F0·V0/4 [9], where V0 represents the theoretical maximum velocity (velocity–axis intercept).”

- Line 214 “All data…..” must be all results.

Response: This has been changed in the manuscript.

- My principal concern relates to the novelty of the study and the clinical value that can be translated for real athletes. Although authors have declared that this is an important step in that direction, however the did not yet explain how can these findings specifically affect some sports individuals performance and training?

Response: We have rewritten the paragraph describing the practical application to make the novelty and clinical value of this study clearer (Track changes version: Lines 352-357; Original version: Lines 324-325):

“Although it is known that phase-specific demands for kinetic characteristics and strength-power capabilities exist in sprint acceleration [22, 26], the physical factors responsible for the sprint acceleration kinetics during specific steps are not clear. The present study revealed the phase-specific demands of physical factors for greater GRFs during sprint acceleration, implying that training interventions tailored to different steps are vital for effectively improving the initial sprint acceleration ability.”

In addition, although it has not been pointed out, the latter part of the same paragraph has been modified as follows at our discretion in order to help the reader image more concretely the training to be performed in the practical field (Track changes version: Lines 361-363; Original version: Lines 329-331). We believe that this clarifies the practical application of the results of this study.

“Thus, heavy-load strength training (e.g., back squats with higher load and lower repetitions) is possibly effective in increasing force production in the earlier phase of the initial sprint acceleration, and explosive strength training (e.g., jump squats with a lighter load) is effective in increasing force production in the later phase.”

- Lines 286-294 : the interpretation of the lower limb strength is a bit confusing and need to be slightly modified for better clarity.

Response: We have revised the points you raised to clarify (Track changes version: Lines 283-300; Original version: Lines 274-293):

“A higher F0 significantly correlated with larger horizontal GRFs for the first, fifth, and ninth steps during sprint acceleration. Theoretically, the amount of the horizontal GRF applied onto the ground is determined by both the magnitude and vector orientation of the resultant GRF. In this study, greater F0 was also associated with a more forward inclination of the resultant GRF vectors for the first and fifth steps, but did not significantly correlate with the magnitude of the resultant GRF. These results imply that the lower-limb maximal strength may influence the direction of the resultant GRF vector rather than its magnitude. During sprint acceleration, a lower resultant GRF might be beneficial for the effective generation of a horizontal GRF, thereby resulting in better sprint acceleration performance [21]. Kugler and Janshen [21] observed that individuals with superior acceleration performance trended toward lower resultant GRF and significantly longer ground contact times, leading to a greater forward lean of the body and therefore greater horizontal force application, particularly during the latter contact phase. In addition, a previous study [22] indicated that during sprint acceleration, a greater increase in sprint velocity was achieved with a small vertical GRF, which is a primary determinant of the magnitude of the resultant GRF, since the magnitude of the vertical GRF required during sprint acceleration might be the one that creates a flight time only long enough to reposition the lower limbs [23]. Therefore, individuals with greater lower-limb strength are likely able to tilt the GRF vector forward while maintaining a certain amount of vertical GRF, and greater lower-limb strength does not necessarily contribute to an increased magnitude of the resultant GRF.”

- The age of the cohort is very limited. I would like to understand the authors perspective about how these results can extend to any other athlete ages.

Response: This is indeed an important point to consider. We believe that our findings are also applicable to other age cohorts of athlete, but that the degree of contribution of maximal strength and power may differ. For example, in the case of youth athletes, greater maximal strength and power of the lower-limb correlated with higher sprint acceleration ability as in adult athletes (Chelly et al. JSCR. 2010; Comfort et al. JSCR. 2014), so it is possible that the greater the maximal strength and power of the lower limb, the larger the GRFs during sprint acceleration is in youth athletes as well. Furthermore, a previous study (Korhonen et al. MSSE. 2009) has reported that, in master athletes, countermovement jump height, which was used as an indicator of force production capacity of the lower limb muscles, was the significant predictor of the mean resultant GRF during propulsive phase of maximal sprint in stepwise regression analysis. Thus, it is considered that lower-limb strength and power contribute to the magnitude of the GRF during sprint running in master athletes as well as the cohort of this study. On the other hand, there are some differences in sprint acceleration kinetics in different age groups. For example, it has been reported that adult athletes showed larger horizontal GRF and ratio of forces (* percentage of horizontal to resultant GRF) for the first step during sprint acceleration than young athletes (Aeles et al. R Soc Open Sci. 2018). Therefore, the magnitude of the correlations might vary across cohorts.

To clarify our thoughts on generalizability to any other athletes, we have revised limitations of this study (Track changes version: Lines 347-349; Original version: Lines 321-322):

“First, the study participants were only male baseball players. It has been suggested that several differences in sprint acceleration kinetics exist between different cohorts, such as those in sport [33], sex [34], and age [35]. Thus, the results might differ if athletes from other sports, female athletes, or youth and master athletes were employed.”

[References]

Aeles J, Jonkers I, Debaere S, Delecluse C, Vanwanseele B. Muscle-tendon unit length changes differ between young and adult sprinters in the first stance phase of sprint running. R Soc Open Sci. 2018;5(6):180332. doi:10.1098/rsos.180332.

Chelly MS, Chérif N, Amar MB, Hermassi S, Fathloun M, Bouhlel E, Tabka Z, Shephard RJ. Relationships of peak leg power, 1 maximal repetition half back squat, and leg muscle volume to 5-m sprint performance of junior soccer players. J Strength Cond Res. 2010;24(1):266-71. doi:10.1519/JSC.0b013e3181c3b298.

Comfort P, Stewart A, Bloom L, Clarkson B. Relationships between strength, sprint, and jump performance in well-trained youth soccer players. J Strength Cond Res. 2014;28(1):173-7. doi:10.1519/JSC.0b013e318291b8c7.

Korhonen MT, Mero AA, Alén M, Sipilä S, Häkkinen K, Liikavainio T, Viitasalo JT, Haverinen MT, Suominen H. Biomechanical and skeletal muscle determinants of maximum running speed with aging. Med Sci Sports Exerc. 2009;41(4):844-56. doi:10.1249/MSS.0b013e3181998366.

Reviewer #2

This paper examines the relationship between the force capacity of the lower limb, estimated by the F-V profile estimation method, and GRF during initial sprint acceleration. Although the paper is novel, there are some parts that seem insufficient in terms of clarity, and there are also some points that require compliance with a policy of self-plagiarism.

- Perhaps, this paper is a series with the abstracts in following proceedings. Since the statistics in the abstract of this paper are almost the same as those listed in the proceedings below, we assume that the main issues are similar. Since I believe there is a discussion accompanying these statistical results in this paper, I assume there are differences from the following proceedings abstract. However, you should cite the following proceedings abstract in this paper and, in accordance with PLOS ONE policy, if necessary I think it is necessary to clarify the difference points.

- "Relationship between leg extension strength characteristics and ground reaction force during sprint acceleration," Motoki Katsuge, Kosuke Hirata, Hikaru Kurosaki, Hiromu Watanabe, Sohma Kambayashi, Kuniaki Hirayama, 2022 SESNZ Conference

Response: You are corr

---

## [Decision Letter · Decision Letter 1]

Dear Dr. Hirayama,

Thank you for submitting your manuscript to PLOS ONE. After careful consideration, we feel that it has merit but does not fully meet PLOS ONE’s publication criteria as it currently stands. Therefore, we invite you to submit a revised version of the manuscript that addresses the points raised during the review process.

**ACADEMIC EDITOR:** Authors provided an improved manuscript, the overall structure is good, however, some concerns are still present and deserve to be addressed in a second round of revision as highlighted by the reviewer.

We look forward to receiving your revised manuscript.

Kind regards,

Andrea Tigrini, Ph.D.

Academic Editor

PLOS ONE

Additional Editor Comments:

Authors provided an improved manuscript, the overall structure is good, however, some concerns are still present and deserve to be addressed in a second round of revision as highlighted by the reviewer.

Reviewers' comments:

Reviewer's Responses to Questions

**Comments to the Author**

Reviewer #2: All comments have been addressed

Reviewer #3: (No Response)

2. Is the manuscript technically sound, and do the data support the conclusions?

Reviewer #2: Yes

Reviewer #3: Partly

3. Has the statistical analysis been performed appropriately and rigorously?

Reviewer #2: Yes

Reviewer #3: Yes

4. Have the authors made all data underlying the findings in their manuscript fully available?

Reviewer #2: Yes

Reviewer #3: Yes

5. Is the manuscript presented in an intelligible fashion and written in standard English?

Reviewer #2: Yes

Reviewer #3: Yes

Reviewer #2: I have carefully reviewed your revised manuscript and response letter. I found that you have adequately addressed all of my points and removed all doubts. I have no further questions.

Reviewer #3: The work done by Hirayama and colleagues focuses on the relationship between lower limb mechanical capabilities and the ground reaction forces from the force-velocity profile for the first, fifth and ninth steps during sprint acceleration. The force-profile by squat jumps has been obtained. Although the paper is well-written in standard technical English, in my opinion, there are some parts that should be improved for a better comprehension from the reader, also the discussion should be carefully revised.

Detailed comments are reported on the attached file "31_PONE-D-24-18473_R1_reviewer_revised".

**Do you want your identity to be public for this peer review?** For information about this choice, including consent withdrawal, please see our Privacy Policy

Reviewer #2: No

Reviewer #3: No

---

## [Author Response · Author response to Decision Letter 2]

28 May 2025

We would like to thank the editor for the time and effort put into the reviewing process of our manuscript and for the opportunity to resubmit the revised version. We have addressed all the points raised by the reviewers with great care.

---

## [Decision Letter · Decision Letter 2]

Relationships between the ground reaction force during initial sprint acceleration and the vertical force–velocity profile

PONE-D-24-18473R2

Dear Dr. Hirayama,

We’re pleased to inform you that your manuscript has been judged scientifically suitable for publication and will be formally accepted for publication once it meets all outstanding technical requirements.

Kind regards,

Andrea Tigrini, Ph.D.

Academic Editor

PLOS ONE

Additional Editor Comments (optional):

Reviewers' comments:

Reviewer's Responses to Questions

**Comments to the Author**

Reviewer #2: All comments have been addressed

Reviewer #3: All comments have been addressed

2. Is the manuscript technically sound, and do the data support the conclusions?

Reviewer #2: Yes

Reviewer #3: Yes

3. Has the statistical analysis been performed appropriately and rigorously?

Reviewer #2: Yes

Reviewer #3: Yes

4. Have the authors made all data underlying the findings in their manuscript fully available?

Reviewer #2: Yes

Reviewer #3: Yes

5. Is the manuscript presented in an intelligible fashion and written in standard English?

Reviewer #2: Yes

Reviewer #3: Yes

Reviewer #2: My concern had already been resolved in the previous version. I have no additional comments regarding the response to my review.

Reviewer #3: (No Response)

**Do you want your identity to be public for this peer review?** For information about this choice, including consent withdrawal, please see our Privacy Policy

Reviewer #2: No

Reviewer #3: No

---

## [Editor Report · Acceptance letter]

PONE-D-24-18473R2

PLOS ONE

Dear Dr. Hirayama,

I'm pleased to inform you that your manuscript has been deemed suitable for publication in PLOS ONE. Congratulations! Your manuscript is now being handed over to our production team.

Kind regards,

on behalf of

Dr. Andrea Tigrini

Academic Editor

PLOS ONE